# Immunogenetic Profile Associated with Patients Living with HIV-1 and Epstein–Barr Virus (EBV) in the Brazilian Amazon Region

**DOI:** 10.3390/v16071012

**Published:** 2024-06-24

**Authors:** Iran Barros Costa, Mayara Natália Santana-da-Silva, Patrícia Yuri Nogami, Carolinne de Jesus Santos e Santos, Leonn Mendes Soares Pereira, Eliane dos Santos França, Amaury Bentes Cunha Freire, Francisco Lúzio de Paula Ramos, Talita Antonia Furtado Monteiro, Olinda Macedo, Rita Catarina Medeiros Sousa, Felipe Bonfim Freitas, Antonio Carlos Rosário Vallinoto, Igor Brasil-Costa

**Affiliations:** 1Epstein-Barr Virus Laboratory, Virology Department, Evandro Chagas Institute, Ananindeua 67030-000, PA, Brazil; irancosta@iec.gov.br (I.B.C.); mnsantanadasilva@gmail.com (M.N.S.-d.-S.); patriciayuri.nogami@gmail.com (P.Y.N.); ennilorac.carol@gmail.com (C.d.J.S.e.S.); elianesantos_85@hotmail.com (E.d.S.F.); talitamonteiro@iec.gov.br (T.A.F.M.); 2Immunology Laboratory, Virology Department, Evandro Chagas Institute, Ananindeua 67030-000, PA, Brazil; 3Virology Laboratory, Institute of Biological Sciences, Federal University of Pará, Belém 66075-110, PA, Brazil; leonnpereira@hotmail.com (L.M.S.P.); vallinoto@ufpa.br (A.C.R.V.); 4Epidemiology and Surveillance Service, Evandro Chagas Institute, Ananindeua 67030-000, PA, Brazil; amaurybentes@iec.gov.br (A.B.C.F.); franciscoramos@iec.gov.br (F.L.d.P.R.); 5Retrovirus Laboratory, Virology Department, Evandro Chagas Institute, Ananindeua 67030-000, PA, Brazil; olindamacedo@iec.gov.br (O.M.); felipebonfim@iec.gov.br (F.B.F.); 6School of Medicine, Federal University of Pará, Belém 66075-110, PA, Brazil; ritaclosset@uol.com.br

**Keywords:** HIV, EBV, polymorphism, single nucleotide, cytokines

## Abstract

Viral coinfection among HIV-positive patients, coupled with the development of AIDS, remains a major public health problem. The synergism between the presence of HIV and other viruses has consequences in relation to changes in the severity of the infection, as well as changes in the natural course of both infections. Several polymorphisms present in genes that encode cytokines have a relevant influence on their transcription and consequently on the production of such immunological molecules. The present study evaluated the influence of SNPs located in the promoter regions of genes encoding the cytokines INF-ɣ, TNF, IL-6, IL-4, and IL-2, as well as their respective plasma concentrations, in patients infected with HIV and/or EBV in the state of Pará. Additionally, this study described the epidemiological profile and compared CD4+ and CD8+ T lymphocyte counts among the groups studied. The associative analysis between the SNPs and plasma cytokine concentrations in different groups showed statistical relevance for three polymorphisms: rs2069762 (*IL2*), where the GG genotype demonstrated higher IL-2 levels in HIV mono-infected individuals; rs2243250 (*IL4*), where the CT genotype showed higher IL-4 levels in the control group; and rs2069705 (*IFNG*), where the TT genotype showed higher IFN-γ levels in the coinfected group. Regarding SNP associations with CD4+/CD8+ counts, significant findings were observed in HIV mono-infected individuals: the rs2069705 (*IFNG*) polymorphism was linked to higher CD4+ counts with the CT genotype, and rs1799964 (*TNF*) was associated with higher CD8+ counts with the CC genotype. Therefore, this study provides evidence that the rs2069705 (*IFNG*) SNP is associated with elevated IFN-γ levels, which may have pathogenic consequences, as depletion of this cytokine is concerning for people living with HIV due to its antiviral properties.

## 1. Introduction

As of June 2023, there were 489,594 reported cases of HIV infection in Brazil, with the northern region ranking fourth with 49,956 cases (10.2%). In 2023, 1311 cases were reported in the state of Pará. Historical data show a significant prevalence of infection in men (70.5%) [1]. In the present study, two groups were composed of individuals infected with HIV, where a statistically significant prevalence of 79% was observed, associated with males in both groups.

Regarding age, the historical series shows a prevalence of 23.4% in the younger age group, between 15 and 24 years, with the highest number of AIDS cases occurring in individuals aged 25 to 39 years, predominantly in males (50.1%) [2]. In the present study, the highest prevalence occurred in the younger age group, between 18 and 29 years, for both groups studied, 50.7% for individuals with HIV mono-infection and 58.6% for those who were HIV/EBV coinfected, suggesting a need for increased attention on public health policies for this population. Epidemiological data from a study conducted in the state of Goiás, in the Central-West region of Brazil, also revealed a higher prevalence of HIV infection in the younger population, with a prevalence of 54.2% (*n* = 539) [3].

The present study addresses the dynamics of infection related to two infectious agents with distinct virological, immunological, and host interaction characteristics. EBV, with a peculiar characteristic in the family to which it belongs, establishes a prolonged period of infection through virological latency alternating with periods of reactivation. On the other hand, HIV establishes a chronic and progressive infection leading to immunodeficiency, a key characteristic of AIDS [4].

The presence of viral coinfection among individuals living with Human Immunodeficiency Virus (HIV), coupled with the development of Acquired Immunodeficiency Syndrome (AIDS), continues to pose a significant global public health challenge. The synergism between HIV and other viruses has implications for changes in the severity of the infection and the natural course of both infections. The occurrence of opportunistic infections is the primary cause of hospitalization and increased morbidity among people living with HIV/AIDS (PLWHA), leading to significant mortality in this population [5,6].

The coinfection of HIV with the Epstein–Barr virus (EBV) constitutes a noteworthy pattern of coinfection, which is associated with significant changes in morbidity and mortality related to HIV [7]. Primary EBV infection gives rise to diverse clinical consequences, ranging from asymptomatic seroconversion (common in children) to infectious mononucleosis (IM) and lymphoproliferative alterations, particularly evident in immunosuppressed patients, such as those coinfected with HIV [7,8].

The establishment of EBV/HIV coinfection modulates the host’s immune system, impacting the number of circulating B lymphocytes infected by EBV, which can vary from 10 to 20 times more in people living with HIV/AIDS (PLWHA) compared to control individuals. This condition can be attributed to the decrease in T lymphocytes caused by HIV, thereby contributing to the lack of control of EBV infection in B cells [9].

Studies assessing the association between the EBV viral load and the development of lymphomas in PLWHA have suggested that elevated viral loads are linked to lymphomas with a more clinically aggressive course, as well as a compromised immune status, as evidenced by lower CD4+ T lymphocyte counts (less than 500 cells per mm^3^). Thus, it is suggested that the EBV viral load is not a predictive factor for the onset of lymphomas in PLWHA but is associated with treatment outcomes for the neoplasm, wherein a decrease in the EBV load is observed [10,11]. Evidence indicates that in patients with IM, the number of CD4+ T lymphocytes specific to EBV constitutes up to 1% of the total circulating cells, decreasing to approximately 0.1% after the resolution of the infection [12].

Therefore, even though they are found in small quantities, it is suggested that specific CD4+ T lymphocytes play a crucial role in controlling EBV infection. This is substantiated when examining individuals clinically diagnosed with AIDS, who exhibit a low count of CD4+ T lymphocytes, leading to high EBV loads and a predisposition to developing neoplasms such as lymphomas [13].

Primary EBV infection in patients with infectious mononucleosis (IM) is characterized by significant elevated levels in the absolute number of CD8+ T lymphocytes in the peripheral blood when compared to asymptomatic patients. Concurrently with the expansion of CD8+ T cells, elevated levels of pro-inflammatory and immunoregulatory cytokines are observed, including interferon-γ (IFN-γ), tumor necrosis factor (TNF), interleukins 6 and 10 (IL-6 and IL-10), and transforming growth factor β (TGF-β) [14,15].

Paradoxically, HIV itself plays a pivotal role in the immune response directed against the infection, facilitated by its own components, such as glycoprotein (gp) 120 and the viral nucleic acids generated during replication. These elements actively stimulate the synthesis of pro-inflammatory cytokines and type 1 interferons, including IFN-α and IFN-β [16].

Several polymorphisms found in genes that encode cytokines exert a significant influence on their transcription, consequently affecting the production of these immunological molecules [17,18,19,20,21]. The selective pressure exerted by various infectious diseases ensures the existence of distinct genotypes in the population associated with cytokine production, some of which prove beneficial in the context of eradicating these diseases. However, by establishing a pro-inflammatory phenotype, there is the potential for the consolidation of a chronic inflammatory condition or progression to a more severe form with a critical clinical outcome for the patient, a situation observed in various infectious diseases [22,23].

Our understanding of the regulatory role of cytokines and the impact of genetic variations on immune response genes is crucial for comprehending the intricate dynamics that influence the outcomes of infectious diseases. In this study, we investigated the relationship between cytokine levels (TNF, IFN-ɣ, IL-6, IL-4, and IL-2), the counts of T CD4+ and CD8+ cells, and the presence of single-nucleotide polymorphisms (SNPs): *IL2*/rs2069762 (-330T/G), *IL4*/rs2243250 (-590C/T), *IL6*/rs1800795 (-174G/C), *IL6*/rs1800796 (-572C/G), *TNF*/rs1799964 (-1031 T/C), and *IFNG*/rs2069705 (-1615C/T). These SNPs are strategically located in the promoter region of genes responsible for cytokine production and immune response dynamics in individuals coinfected with HIV and EBV.

## 2. Materials and Methods

### 2.1. Sampling

This study encompassed a cohort of 655 individuals sourced from patients receiving care at prominent public institutions: Instituto Evandro Chagas (IEC), CASA DIA (Center for Health Care in Acquired Infectious Diseases—Department of STIs, AIDS, and Viral Hepatitis), and the Center for Hemotherapy and Hematology of the State of Pará (HEMOPA). The patient population was stratified into four groups: Group 1 (G1)—individuals with positive serology for active HIV and EBV infection (*n* = 79); Group 2 (G2)—individuals with positive serology for active HIV and negative for EBV (*n* = 322); Group 3 (G3)—individuals with positive serology (IgM) for active EBV infection and negative for HIV (*n* = 34); and Group 4 (G4)—the control group, consisting of 220 samples from healthy individuals with negative serology for both HIV and EBV. HIV-positive individuals were ART-free.

Peripheral whole blood samples were collected from male and female individuals aged 18 years or older who had not used medications for HIV (up until the sample collection phase, the study population had not yet begun antiretroviral therapy (ART)) or EBV infection treatment. The acquisition of epidemiological characteristics, such as gender, age, education, and others, relied on the application and subsequent analysis of an epidemiological questionnaire. All participants were informed about the research objectives and willingly provided their consent by signing a consent form. This study was conducted in accordance with the Declaration of Helsinki and received approval from the Research Ethics Committee of Instituto Evandro Chagas (CAAE 73927717.3.0000.0019 CEP-IEC). To uphold participant privacy, all analyzed data were anonymized.

### 2.2. Serological Methods

The blood samples were centrifuged (12298 RCF) for 10 min, followed by separation of the plasma from the cellular mass (buffy coat). Subsequently, the samples were stored in an ultrafreezer at −70 °C until their use.

The obtained plasmas were then employed in the semi-quantitative analysis of IgM and IgG anti-VCA antibodies (EBV). For this analysis, an ELISA (RIDASCREEN EBV VCA, R-BIOPHARM, Darmstadt, Hesse, Germany) was utilized, following the manufacturer’s instructions. Concerning the diagnosis of HIV infection, the qualitative detection of IgG anti-p24 antibodies was conducted using the fourth-generation ELISA method (MUREX AG/AB COMBINATION, DIASORIN, Dartford, Kent, UK) as an initial screening. To confirm HIV infection, the rapid immunoblot test DPP HIV-1/2 (BIOMANGUINHOS, FIOCRUZ, Rio de Janeiro, RJ, Brazil) was applied.

### 2.3. Plasma Measurement of Cytokine Levels

The plasma concentrations of cytokines were determined by employing flow cytometry and the BDTM Cytometric Bead Array (CBA) Human Th1/Th2/Th17 Cytokine Kit, following the manufacturer’s guidelines (Becton & Dickinson, San Jose, CA, USA). Measurements were conducted using the FACSCanto™ II cytometer (Becton & Dickinson, CA, USA). The plasma levels were expressed in pg/mL, and the cytokine detection limits were set as follows: IL-2 (2.6 pg/mL), IL-4 (4.9 pg/mL), IL-6 (2.4 pg/mL), TNF (3.8 pg/mL), and IFN-ɣ (3.7 pg/mL). Subsequent to sample acquisition by the cytometer, the resulting data were exported to the FCAP Array^TM^ software version 3.0 (Becton & Dickinson, San Jose, CA, USA) to conclude the cytokine measurement process.

### 2.4. Cell Counting

Within a period of up to 72 h following peripheral blood collection, the counting of CD4+ T and CD8+ T lymphocytes was carried out using flow cytometry. This procedure was conducted by employing the BDFACSCalibur equipment in conjunction with the monitoring kits BD FACSCalibur, TriTEST™/TruCount, and ACSCount^TM^ Reagents (BD Biosciences, San Jose, CA, USA).

### 2.5. Nucleic Acid Extraction

The buffy coat samples were subjected to DNA extraction at the IEC, using the QIAamp DNA Mini Kit (Qiagen, Düsseldorf, Nordrhein-Westfalen, Germany), following the recommendations for the protocol defined by the manufacturer, for subsequent research into SNPs.

### 2.6. Quantification of HIV Viral Load

To determine the HIV plasma viral load, the HIV-1 Viral Load Amplification Kit (ABBOTT Molecular, Des Plaines, IL, USA) was employed. Viral load quantifications were reported in both copies/mL and log10.

### 2.7. Selection of SNPs and Genotyping

The SNPs were selected based on association studies found in the PubMed database (www.ncbi.nlm.nih.gov/, accessed on 1 January 2022). An investigated SNP was chosen based on criteria such as its location in the promoter region of the genes, playing a role in the transcriptional regulation of the cytokine. Transcription factors bind to this region, influencing its activity and the prevalence of the polymorphism. Table 1 summarizes the main features related to the polymorphisms investigated in the present work.

SNPs were investigated using the hydrolysis probe genotyping methodology for each allele, which is based on the use of an RNase H2 responsible for activating the primers after efficient binding to their target sites, reducing the formation of primer dimers and improving the specificity of the reaction. The rhAmp™ SNP Genotyping kit was used, following the manufacturer’s recommendations (Integrated DNA Technologies, Coralville, IA, USA).

### 2.8. Statistical Analysis

Differences in demographic and clinical characteristics were assessed using the Kruskal–Wallis test for non-parametric quantitative variables and the chi-squared test for qualitative variables. Allelic and genotypic frequencies corresponding to cytokine gene promoter regions were examined for Hardy–Weinberg equilibrium and compared among the four groups using the Kruskal–Wallis test for non-parametric variables and ANOVA for parametric variables, with covariate adjustment. The analyses were performed using JASP version 0.18.3 [24] and Bioestat 5.3 [25]. Statistical significance was considered at *p* ≤ 0.05.

## 3. Results

### 3.1. Demographic Characteristics of the Study Population

This study comprised a population of 655 individuals distributed across four groups based on their serological statuses for the investigated infectious agents, EBV and HIV. Table 2 summarizes the demographic characteristics of the four studied groups. We observed significant age differences among the groups (*p* < 0.01), with G2 (HIV+/EBV−) composed of younger individuals and G4 (control) consisting of older individuals. Regarding the sex variable, there was a predominance of males, with significant differences except for G3 (EBV+/HIV−).

The analysis of sexual orientation revealed significant differences between groups (*p* < 0.01), with a higher frequency of the homosexual population in groups G1 (EBV+/HIV+) and G2 (HIV+/EBV−), and of the heterosexual category in groups G3 and G4. Regarding marital status, there was a higher prevalence of single individuals (67.63%), followed by married individuals (26.57%), with a statistically significant difference between the categories analyzed (*p* < 0.01). The analysis between groups showed the same pattern, with the exception of group G4, which presented the highest number of married individuals (Table 2).

The variable educational level showed a higher frequency of individuals with higher education (42.75%), followed by secondary education (41.07%) and elementary education (15.72%). When the analysis was directed by groups, it was evidenced that in groups G1, G2, and G3 there was a higher frequency of individuals with secondary education. Differences were demonstrated between the categories of elementary, secondary, and higher education, which presented the greatest statistical discrepancy in relation to groups G2, G3, and G4, respectively (Table 2). The analysis of household income showed that the majority of the study population belonged to lower-income groups. There was a significant difference in income distribution between the three categories (<1× minimum wage, 1–3× minimum wage, and >3× minimum wage). The greatest discrepancies were observed between groups G1, G2, and G4, respectively (Table 2).

### 3.2. Analysis of CD4+ and CD8+ T Lymphocyte Counts among the Study Groups

The comparison between groups regarding the counts of T CD4+ and CD8+ lymphocytes revealed significant differences. Concerning the CD4+ T lymphocyte count, disparities were observed in all comparisons made in relation to the control group: coinfected versus control (*p* < 0.01) (Figure 1A), HIV+/EBV− versus control (*p* < 0.01) (Figure 1B), and HIV−/EBV+ versus control (*p* < 0.05) (Figure 1C). Additionally, significant differences were found between the coinfected group (Figure 1E) and those who were EBV mono-infected (*p* < 0.01) (Figure 1D) and between individuals with HIV mono-infection and those who were EBV mono-infected (*p* < 0.01) groups (Figure 1F).

Similarly to our observations for the CD4+ T lymphocyte count, significant differences were found in all comparisons performed regarding the control group with CD8+ T lymphocytes. The latter consistently showed lower mean values in all analyses: coinfected versus control group (*p* < 0.01) (Figure 2A), HIV+/EBV− versus control (*p* < 0.01) (Figure 2B), and HIV−/EBV+ versus control (*p* < 0.01) (Figure 2C). In addition to comparisons with the control group, differences between coinfected and EBV mono-infected groups (*p* < 0.05) (Figure 2D), and coinfected and HIV mono-infected groups (*p* < 0.01) were noteworthy (Figure 2E). There was no significant difference between the individuals with HIV mono-infection and those with EBV mono-infection (Figure 2F).

### 3.3. SNPs and Plasma Concentrations of Cytokines IL-2, IL-4, IL-6, TNF-α, and IFN-γ

The genotype frequency observed was not different between the groups (*p* > 0.05). A comparative analysis was carried out between the genotypes of different SNPs and the plasma concentrations of the cytokines IL-2, IL-4, IL-6, TNF-α, and IFN-γ. The analysis between the different SNPs and the plasma concentration of cytokines showed an association only for three SNPs: rs2069762 (*IL2*), rs2243250 (*IL4*), and rs2069705 (*IFNG*).

In relation to rs2069762 (IL-2), there was a significant difference between the homozygous GG genotype and the heterozygous TG and homozygous TT genotypes in relation to those mono-infected with HIV (G2), where there were higher plasma levels of IL-2 related to the GG genotype (Figure 3). In relation to rs2243250 (*IL4*), there was a significant difference between the CC homozygous genotype and the CT heterozygous genotype in relation to the control group (G4), where higher plasma levels of IL-4 were associated with the heterozygous genotype. (Figure 4). We also observed rs2069705 to have significant relevance (*IFNG*). The difference occurred in G4, where it was observed that the homozygous TT genotype expressed higher plasma levels of IFN-γ when compared with the other genotypes (Figure 5). The other analyses between the SNPs evaluated and the plasma concentrations of the cytokines are not included in the text of this article as they did not show any statistically significant difference, but they are grouped and included in the Appendix A, as described in Appendix A.

### 3.4. SNPs and CD4+ and CD8+ Lymphocyte Counts

Concerning the analysis of the CD4+ T lymphocyte count, there was an association only for rs2069705 (*IFNG*) directed to the group of individuals with HIV mono-infection (G2) (Figure 6). A statistically significant difference was observed between the homozygous CC genotype and the heterozygous CT genotype, which showed higher cell counts associated with the individuals with HIV mono-infection (G2).

The analysis between the different SNPs and the CD8+ T lymphocyte count revealed a statistical association with three SNPs: rs1799964 (*TNF*), rs2069705 (*IFNG*), and rs1800796 (*IL6*). The HIV+/EBV- group (G2) demonstrated a significant difference for the rs1799964 SNP (*p* < 0.05), where the homozygous CC genotype was related to higher counts of CD8+ T cells (Figure 7).

Regarding the control group (G4), statistically significant differences were observed for two SNPs: rs1800796 (*IL6*) and rs2069705 (*IFNG*). The rs1800796 (*IL-6*) SNP showed a higher quantity of CD8+ in individuals with the homozygous CC genotype, as depicted in Figure 8. Concerning the rs2069705 (*IFNG*) SNP, there was statistical relevance regarding the homozygous CC and heterozygous CT genotypes, the latter being associated with higher counts of cytotoxic T cells (Figure 9).

The other analyses between the SNPs evaluated and the CD4+/CD8+ lymphocyte count are also not included in the text of this article because they did not show a statistically significant difference, but they are also grouped together and included in the Appendix A, as described in Appendix A.

### 3.5. SNPs and HIV Viral Load

The final analysis determined the relationship between the SNPs in this study and the viral loads of individuals with HIV/EBV coinfection (G1) and individuals with HIV mono-infection (G2). Although no statistically significant results were found in this analysis, the complete data can be found in Appendix A for inspection and further analysis.

## 4. Discussion

Considering the significance of the antiviral response facilitated by acquired immunity through CD4+ and CD8+ T lymphocyte counts, this study undertook a comparative analysis among different groups, with a focus on T-cell counts [26]. Disparities in T-cell counts, specifically CD4+ and CD8+ T lymphocytes, were evident among the groups. Notably, the coinfected group displayed lower CD4+ T-cell counts when juxtaposed with the other groups, a trend attributed to the presence of HIV infection.

Some differences were evident in relation to cell-mediated adaptive immunity, where we observed lower counts of CD4+ T cells associated with HIV-infected individuals and the reverse occurring in relation to CD8+ T cells, which showed higher counts associated with HIV-infected individuals

EBV is directly associated with the development of lymphoproliferative diseases, including AIDS-related lymphomas, such as Burkitt’s lymphoma, as well as Hodgkin’s lymphoma related to HIV infection [27]. A study has shown that the relative frequency of regulatory T cells (Tregs) increases in acute HIV infection, but during the treatment of chronic HIV infection, the number of regulatory cells is restored, suggesting that uncontrolled HIV infection may contribute to the immune escape of EBV [28].

A study comparing individuals with EBV/HIV coinfection with healthy patients serologically reactive for EBV showed the presence of decreased immune surveillance in relation to EBV-specific CD4+ and CD8+ T cells in HIV-positive individuals, with exacerbated immunological alterations in individuals at a more advanced stage of the disease, such as a loss in the production of TNF-α, IFN-γ, and IL-2 by subpopulations of CD4+ T cells and a reduction in memory effector CD8+ T cells [29].

Regarding CD4+ T lymphocyte counts, notable disparities were observed in all comparisons against the control group: coinfected versus control, HIV+/EBV− versus control, and HIV-/EBV+ versus control. Additionally, significant differences were observed for the coinfected group versus those mono-infected with EBV and for individuals with HIV mono-infection versus those mono-infected with EBV. These could explain the observed lower CD4+ T lymphocyte counts in the HIV-infected groups. Furthermore, this situation may have implications for the production of IFN-γ, a crucial cytokine in viral control [30,31]. It is also known that in HIV-1 infection, there is an initial response through the activation and recruitment of CD4+ T cells, as occurs in the presence of EBV infection during infectious mononucleosis [32]. This may explain the higher CD4+ T-cell counts in the EBV mono-infected group when compared to the HIV co- and mono-infected groups in this study.

Concerning the CD8+ T-cell count, a study conducted on individuals with HIV infection assessed the count of CD8+ T cells before and after prolonged use of antiretroviral therapy (ART). The findings indicated a noteworthy increase in cell counts in individuals not yet under ART, followed by a subsequent decrease after the commencement of treatment [33]. This pattern aligns with the conclusions drawn in this study, showcasing a noteworthy count of CD8+ T cells in the HIV-associated groups without ART usage.

An important component of the host’s immune response against viral infections was evaluated, that is, the regulation of cytokine production. Fluctuation in the expression of these plasma proteins can positively or negatively impact disease progression over time, making them clinically relevant parameters for patient assessment. Studies with different infectious agents have demonstrated increased levels of inflammatory cytokines, such as IFN-γ and TNF-α, after infection is established [34,35,36,37].

Plasma levels of the cytokine IFN-γ revealed that the case groups (mono- and coinfected) expressed, on average, higher plasma concentrations compared to the control group, a fact possibly justified by the anti-inflammatory characteristic of the antiviral response [38]. Similar data were demonstrated by Silva et al. [39] related to HIV/HHV8 coinfection and individuals with HIV mono-infection. However, slightly lower mean concentrations were observed in these groups compared to those mono-infected with EBV. This can be explained by a potential depletion of CD4+ T lymphocytes imposed by the presence of HIV, since the study population was not on antiretroviral therapy at the time of sample collection.

It is worth noting that IFN-γ has a potent antiviral property; thus, its production is crucial for PLHIV to maintain a minimum balance in the immune response, ensuring a more positive prognosis. However, the coinfection scenario with EBV, expressing genes such as *BZLF1*, may compromise this balance in favor of an immune escape mechanism [40]. SNPs located in gene promoter regions can be directly associated with a differential modulation of transcriptional levels, imposing a positive or negative regulation of plasma cytokine expression, a process that can impact the course of various infections.

The *IFNG* gene is located on chromosome 12q14, with several SNPs in this gene reported and associated with diseases such as aplastic anemia, infectious hepatitis, systemic lupus erythematosus, and asthma [41,42,43,44,45]. In the context of SNPs and plasma levels of cytokines, there was a significant association related to the SNP rs2069705 (IFNG) between the homozygous TT genotype and the other genotypes, homozygous CC and heterozygous CT, in relation to the control group, where individuals with the homozygous TT genotype expressed higher levels of IFN-γ. A study aiming to analyze the relationship between the SNP rs2069705 *(IFNG*) and susceptibility to severe malaria in pregnant women categorized three groups: mild malaria, severe malaria, and the control group. Regarding allelic and genotypic frequency analysis, they found a higher prevalence of the homozygous TT genotype (66.2%) and the T allele (88%) associated with more severe cases of malaria in this population. This was also correlated with higher IFN-γ concentrations when compared with the heterozygous TC genotype. No influence of the homozygous CC genotype was found on the plasma IFN-γ concentration [46].

As for the correlation between the genotype and plasma IFN-γ concentration, there is a disparity between their findings and ours, as the authors reported elevated concentrations associated with the homozygous TT genotype. The discrepancy can be explained by the analyses’ focuses on genetically distinct populations, as the population admixture and genetic heterogeneity can cause false associations between genetic polymorphisms and diseases. It has been described that the SNP rs2069705 (*IFNG*) differs significantly between Chinese and white individuals, as was highlighted in a study that evaluated the association of IFN-γ-related polymorphisms with breast cancer in healthy cases and controls, with frequencies of 50.9% and 52.9% for the homozygous GG genotype and 8.4% and 6.3% for the homozygous TT genotype, respectively, within a Chinese population, which aligns with the present study [17].

A case–control study conducted in the United States on women with and without breast cancer showed genotypic frequencies for SNP rs2069705 that were discordant with the findings of the present study. In a predominantly white population, genotypic frequencies of 20%, 43.3%, and 36.7% were observed for the CC, CT, and TT genotypes, respectively, with no statistical association between plasma IFN-γ levels and the SNP rs2069705, a fact that can be explained by the low sample number of the evaluated population (*n* = 40). In sum, multiple scenarios concerning the respective genotypes can be observed [47]. The SNP rs2069705 (*IL6*) is also associated with inflammatory diseases such as systemic lupus erythematosus, characterized by high IFN-γ concentrations, in which it influences the pathogenesis of the diseases [38].

A case–control study sought to confirm the relationship between SNP rs2069762 (-330 T/G) and patients with chronic thrombocytopenia. The results showed that patients with the disease had high levels of IL-2, conferring an inflammatory character to the immune response by presenting potent lymphoproliferative effects, thus inducing a strong expansion of Th1 cells and a consequent increase in IFN-γ production. Another relevant finding of the study concerned the genotype frequencies found in the control group of blood donors, which were similar to those found in this study [48].

A further point of analysis in the present study concerned the association between SNPs and the CD4+ and CD8+ T-cell counts. A significant association was observed between SNP rs2069705 and the CD4+ T-cell count in the individuals with HIV mono-infection. The CC genotype was associated with lower CD4+ T-cell counts in individuals with HIV mono-infection. According to the Centers for Disease Control and Prevention [49], CD4+ T-cell counts of less than 200 cells/mm^3^ are categorized as stage 3 HIV infection, enhancing the immunodeficiency condition. These results suggest that the CC genotype of rs2069705 is related to the decrease in immunological effector and regulatory functions, leading to immunodeficiency conditions [50].

It is known that the production of IL-4 is directly linked to the genetic characteristics of the SNP rs2243250 (-590 C/T), which is located 589 bp upstream of the transcription site, and that the transition from the C allele to the T allele is associated with an increase in the expression of the gene, as well as the establishment of stronger interactions between the transcriptional factors and its promoter, consequently altering the expression levels of the IL-4 protein [51].

It is also known that IL-4 contributes significantly to the modulation of a potent anti-inflammatory response. Decreased concentrations of IL-4 have been documented in patients with rheumatoid arthritis who express the TT genotype for the rs2243250 SNP, suggesting that the T allele may function as a biomarker for the intensely inflammatory disease. However, it has already been described that individuals expressing the C allele of the rs2243250 SNP show reduced susceptibility to rheumatoid arthritis, with it acting as a protective factor [52].

As for CD8+ T cells, associations were identified with five SNPs: rs2243250, rs1800795, rs1800796, rs1799964, and rs2069705. Notably, homozygous CC genotypes for rs1800796 (*IL6*) were associated with higher CD8+ T cells in individuals from the control group. In addition, homozygous CC genotypes for rs1799964 (TNF) were associated with higher CD8+ and lower CD4+ T-cell counts compared to some individuals with HIV mono-infection group. These results suggest that SNP genotypes are related to more active cytotoxic responses in relation to HIV and/or EBV infections. It was observed by Sinha and collaborators that the rs2281808 TT SNP in the *SIRPG* gene is associated with the expressive effector state of CD8+ T lymphocytes in autoimmune disease, despite the reduction in absolute count [53].

This study had potential limitations. The cohort was notably heterogeneous due to the method of sample acquisition, which was based on spontaneous demand. Moreover, the samples were collected from patients who had recently been diagnosed with the infection and were subsequently referred to a reference center for appropriate treatment. Furthermore, we chose to evaluate samples free of antiretroviral therapy (ART) to analyze the progression of the infection without the confounding effects of ART, as our objective was to examine the influence of the host’s immune response to the infection.

## 5. Conclusions

Given the present scenario for the socio-epidemiological profile of individuals living with HIV, there is an evident need for public policies adapted to young populations. HIV coinfection can lead to EBV reactivation, consequently increasing plasma concentrations of IFN-γ and TNF-α. With regard to CD4+ T-cell counts, the significant decrease in counts associated with the group of HIV-infected individuals not undergoing ART, associated with the CC genotype of SNP rs2069705, when compared to the control group and those infected only with EBV clearly shows the different modulation of the immune system imposed by the two viruses, which have different replicative characteristics, where HIV leads to considerable elimination of CD4+ T cells. The opposite was observed for CD8+ T cells, with high counts, suggesting a more active cytotoxic response and acute infection, characteristic of HIV-infected and treatment-free patients, where CD8+ proliferation is observed in the peripheral blood with a subsequent decline associated with the establishment of ART. Therefore, monitoring the CD8+ T-cell count can function as a biological marker of disease progression.

This study provides evidence that the TT genotype of SNP rs2069705 (*IFN*) is associated with high IFN-γ levels and that this may have impacts on the viral immune response. Higher levels of IL-2 were associated with the homozygous GG genotype of SNP rs2069762 in HIV-infected individuals.

## Figures and Tables

**Figure 1 viruses-16-01012-f001:**
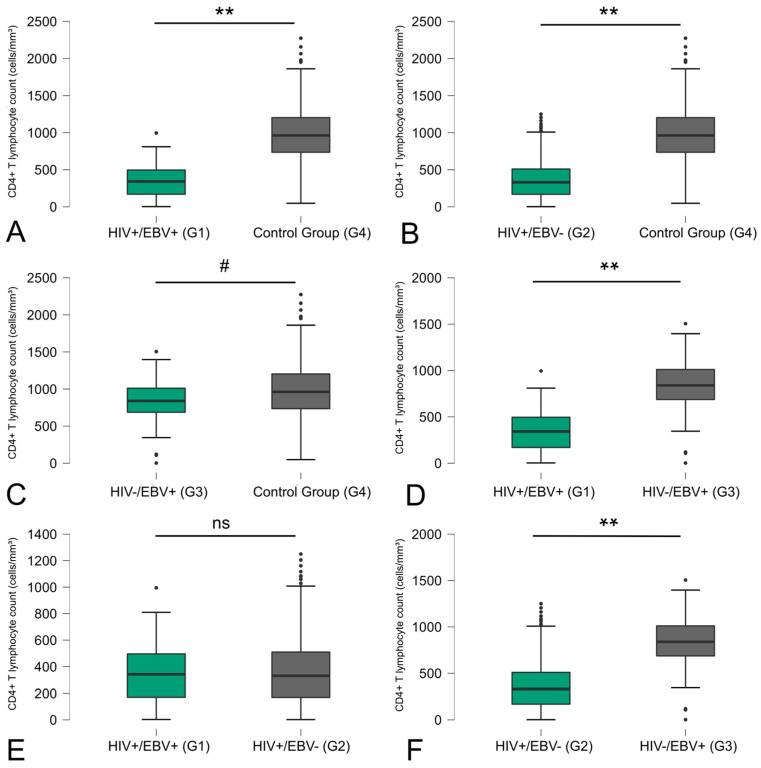
Comparative distribution of CD4+ T-cell counts between the study groups. The *p*-values are calculated using the Kruskal–Wallis test and are adjusted for multiple comparisons using the Bonferroni correction. ** *p* < 0.01; # *p* < 0.05; ns: not significant.

**Figure 2 viruses-16-01012-f002:**
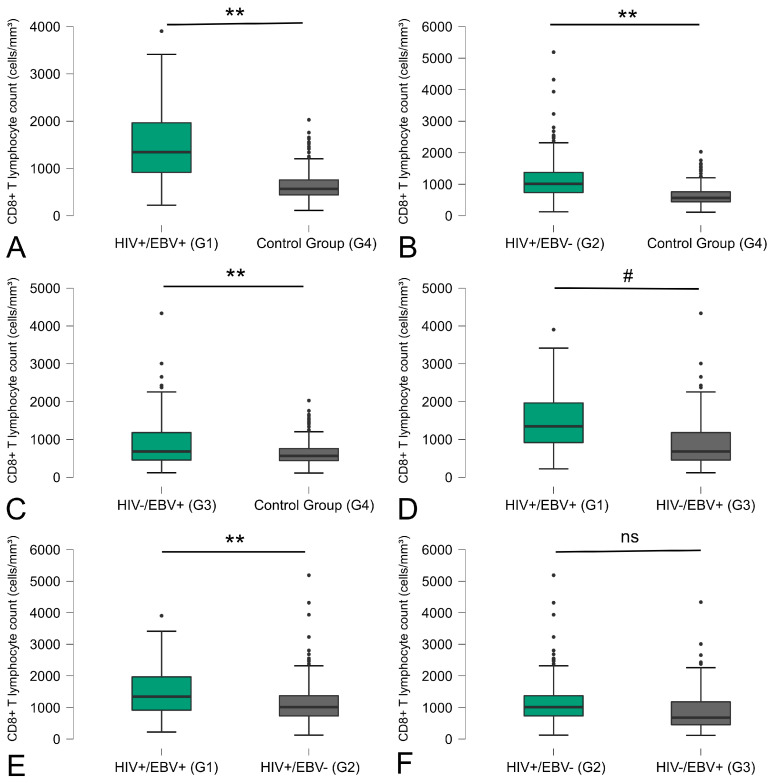
Comparative distribution of CD8+ T-cell counts between the study groups. The *p*-values are calculated using the Kruskal–Wallis test and are adjusted for multiple comparisons using the Bonferroni correction. ** *p* < 0.01; # *p* < 0.05; ns: not significant.

**Figure 3 viruses-16-01012-f003:**
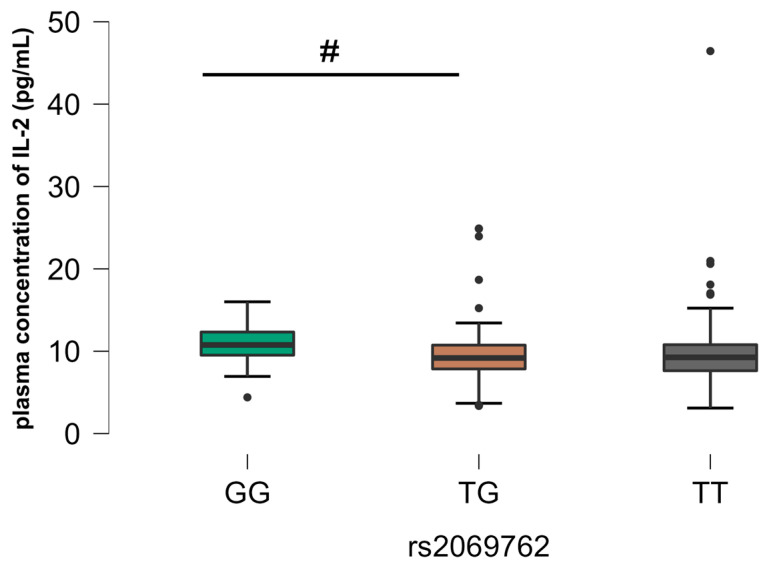
Comparative analysis between SNP rs2069762 and plasma IL-2 concentration, group G2. The *p*-values are calculated using the Kruskal–Wallis test and are adjusted for multiple comparisons using the Bonferroni correction. # *p* < 0.05.

**Figure 4 viruses-16-01012-f004:**
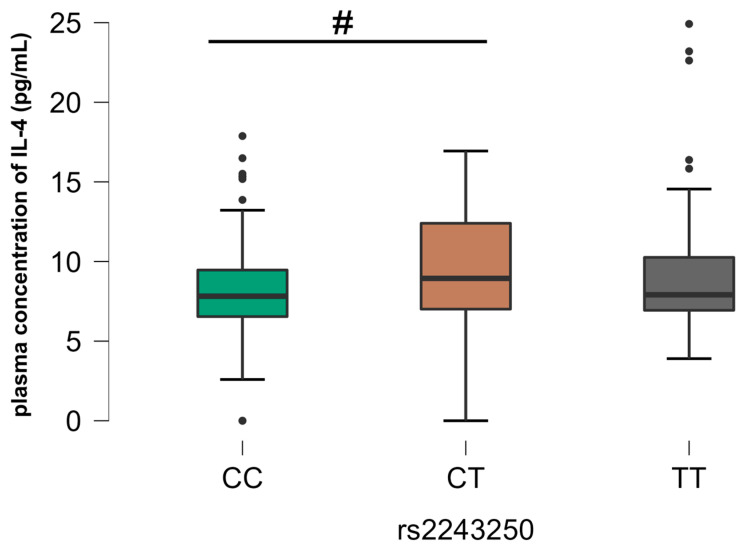
Comparative analysis between SNP rs2243250 and plasma IL-4 concentration, group G4. The *p*-values are calculated using the Kruskal–Wallis test and are adjusted for multiple comparisons using the Bonferroni correction. # *p* < 0.05.

**Figure 5 viruses-16-01012-f005:**
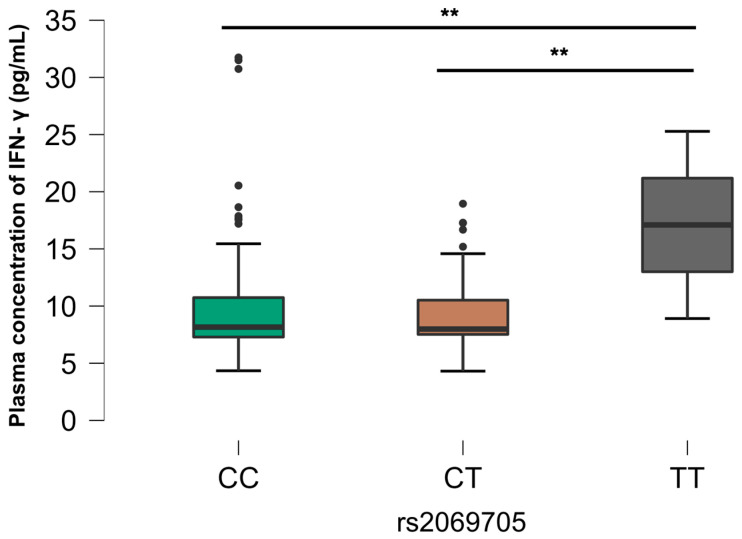
Comparative analysis between SNP rs2069705 and plasma IFN-y concentration, group G4. The *p*-values are calculated using the Kruskal–Wallis test and are adjusted for multiple comparisons using the Bonferroni correction. ** *p* < 0.01.

**Figure 6 viruses-16-01012-f006:**
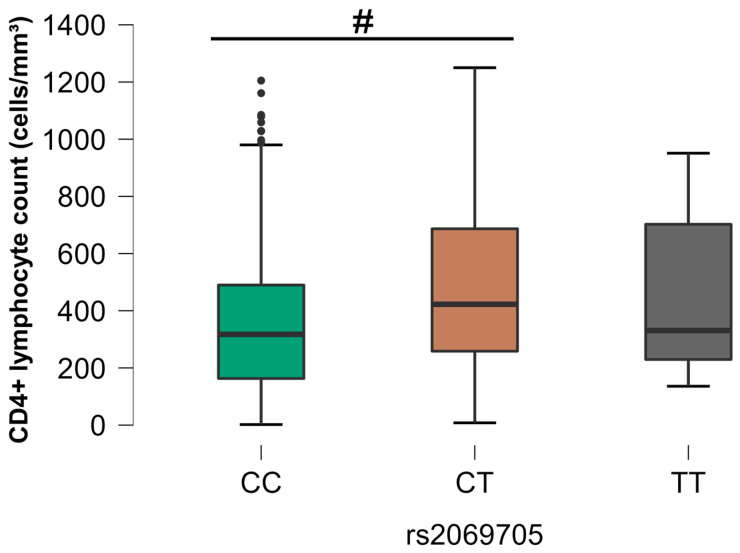
Comparative analysis between SNP rs2069705 and CD4+ T lymphocyte count, group G2. The *p*-values are calculated using the ANOVA test and are adjusted for multiple comparisons using the Bonferroni correction. # *p* < 0.05.

**Figure 7 viruses-16-01012-f007:**
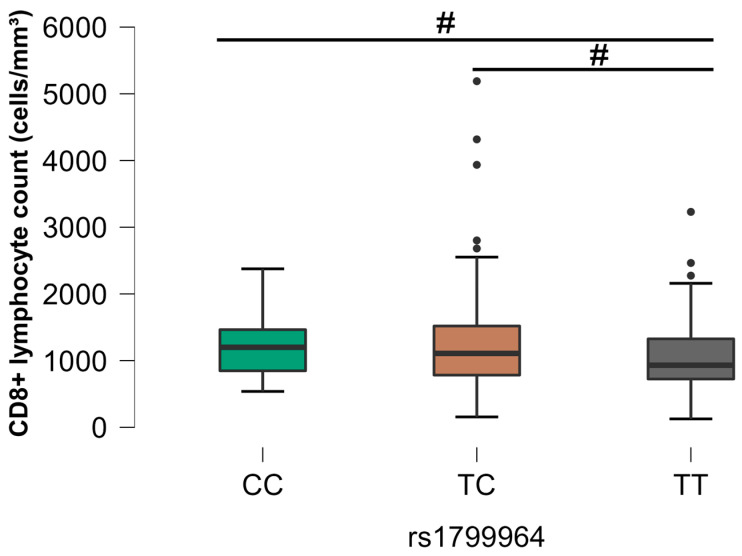
Comparative analysis between the SNP rs1799964 (*TNF*) and the CD8+ T lymphocyte count, group G2. The *p*-values are calculated using the Kruskal–Wallis test and are adjusted for multiple comparisons using the Bonferroni correction. # *p* < 0.05.

**Figure 8 viruses-16-01012-f008:**
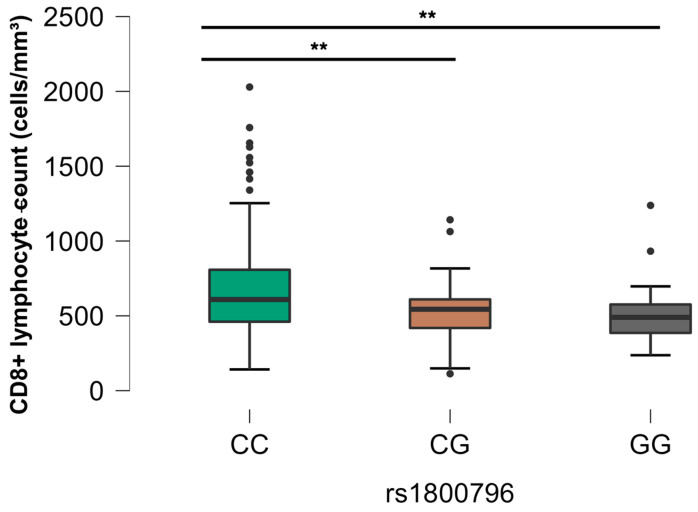
Comparative analysis between SNP rs1800796 (*IL6*) and CD8+ T lymphocyte count, control group (G4). The *p*-values are calculated using the Kruskal–Wallis test and are adjusted for multiple comparisons using the Bonferroni correction. ** *p* < 0.01.

**Figure 9 viruses-16-01012-f009:**
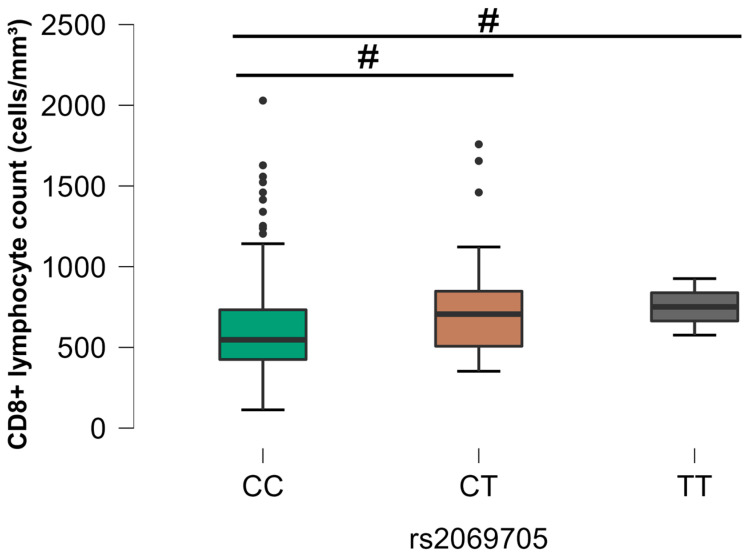
Comparative analysis between SNP rs2069705 (*IFNG*) and CD8+ T lymphocyte count, control group (G4). The *p*-values are calculated using the Kruskal–Wallis test and are adjusted for multiple comparisons using the Bonferroni correction. # *p* < 0.05.

**Table 1 viruses-16-01012-t001:** Characterization of the polymorphisms studied.

Gene	ID	Allele	Function	MAF *
*IL2*	rs2069762	T/G	Upstream Variant	0.50
*IL4*	rs2243250	C/T	Upstream Variant	0.46
*IL6*	rs1800795	G/C	Intron Variant	0.14
*IL6*	rs1800796	C/G	Upstream Variant	0.30
*TNF*	rs1799964	T/C	Upstream Variant	0.21
*IFNG*	rs2069705	C/T	Upstream Variant	0.48

* Minor allele frequency according 1000 Genomes Project.

**Table 2 viruses-16-01012-t002:** Demographic characteristics of the four studied groups: individuals with positive serology for active HIV and EBV infection (G1), individuals with positive serology for active HIV and negative for EBV (G2); individuals with positive serology for active EBV infection and negative for HIV (G3); and the control group (G4).

Variable	G1 (*n* = 79)	G2 (*n* = 322)	G3 (*n* = 34)	G4 (*n* = 220)	*p*-Value
**Age (Median)**	29	27	32	35	<0.01 ^1^
**Sex [*n* (%)]**					
Male	63 (79.7%)	256 (79.5%)	17 (50%)	141 (64.1%)	<0.01 ^2^
Female	16 (20.3%)	66 (20.5%)	17 (50%)	79 (35.9%)	
**Sexual orientation [*n* (%)]**					
Homosexual	35 (44.4%)	142 (44.2%)	3 (8.8%)	3 (1.3%)	<0.01 ^2^
Bisexual	15 (18.9%)	47 (14.6%)	0 (0.0%)	3 (1.3%)	
Heterosexual	29 (36.7%)	132 (40.9%)	30 (88.3%)	210 (95.4%)	
Not reported	0 (0.0%)	1 (0.3%)	1 (2.9%)	4 (1.8%)	
**Marital status [*n* (%)]**					
Single	70 (88.6%)	253 (78.6%)	23 (67.6%)	97 (44.1%)	<0.01 ^2^
Married	6 (7.6%)	55 (17.1%)	9 (26.5%	104 (47.3%)	
Divorced	2 (2.5%)	7 (2.1%)	2 (5.9%)	12 (5.4%)	
Widowed	0 (0.0%)	6 (1.8%)	0 (0.0%)	4 (1.8%)	
Not reported	1 (1.3%)	1 (0.3%)	0 (0.0%)	3 (1.4%)	
**Educational level [*n* (%)]**					
Elementary school	15 (18.99%)	62 (19.25%)	4 (11.76%)	22 (10.0%)	<0.01 ^2^
High school	40 (50.63%)	140 (43.48%)	16 (47.06%)	73 (33.19%)	
College/University	24 (30.38%)	120 (37.27%)	14 (41.18%)	122 (55.45%)	
Illiterate	0 (0.0%)	0 (0.0%)	0 (0.0%)	0 (0.0%)	
Not reported	0 (0.0%)	0 (0.0%)	0 (0.0%)	3 (1.36%)	
**Household income [*n* (%)]**					
Less than 1× minimum wage	20 (25.3%)	54 (16.8%)	2 (5.9%)	29 (13.2%)	<0.01 ^2^
1–3× minimum wage	48 (60.8%)	224 (69.5%)	21 (61.7%)	130 (59.1%)	
More than 3× minimum wage	5 (6.3%)	25 (7.8%)	8 (23.5%)	55 (25%)	
Not reported	1 (1.3%)	0 (0.0%)	0 (0.0%)	0 (0.0%)	
Do not know	5 (6.3%)	19 (5.9%)	3 (8.9%)	6 (2.7%)	

Categorical variables are expressed as percentages. Age value is expressed as median. ^1^ *p*-values were calculated by Kruskal–Wallis test; ^2^ *p*-values were calculated by chi-squared test.

## Data Availability

Data supporting the findings of this study are available from Igor Brasil Costa, but restrictions apply to the availability of these data, which were used under license for the present study and therefore are not publicly available. However, data are available from the authors upon reasonable request and with permission from Igor Brasil Costa.

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
