# Peer review of "Immunogenetic Profile Associated with Patients Living with HIV-1 and Epstein–Barr Virus (EBV) in the Brazilian Amazon Region"

_viruses, 2024, doi:10.3390/v16071012_

Round 1

Reviewer 1 Report

Comments and Suggestions for Authors

The study deals with polymorphic variants of some cytokine-encoding genes predisposing for distinct clinical course of HIV infection being accomplished by detectable Epstein-Barr virus (EBV). This item is of sufficient clinical value and presents certain novel data concerning HIV/EBV epidemiology in regional population.

Remarks. Abstract:

Line 21: The distinct aim of the study is not presented, description of clinical and laboratory parameters is listed instead.

The authors declare presence of differences …in all comparisons made in relation to the control group (G4)…. However, no results are present concerning potential differences except of expected differences, e.g., in CD4+ cells against control group. Among infected groups, only a difference of co-infected vs. mono-infected EBV is reported (for which parameters?). The results of SNP- immunity associations are not found, neither in Abstract, nor in Conclusion.

Materials and methods

Line 103-105 and Table 1. One should provide data on mean (mediane) terms of HIV infection (years)  in  HIV patients, or duration of ARVT in these groups, because ARVT is a sufficient factor of immune changes as noted further in Discussion. The ARVT effects may sufficiently hide the genetic associations with SNP’s studied.

Line 156: SNPs genotyping: was it performed by original technique (e.g., primer design), or by commercial test system as mentioned (line 168).

Results:

Section 3.3: (SNPs and cytokine concentrations (line 241 and further on): The text description looks non-systematic and fragmentary, with a series of figures presenting some changes of different plasma cytokine contents in different study groups with SNP’s of appropriate cytokine genes. This design prevents demonstration of clear associations between SNP’s and probable immune effects (upon CD4+ cells and cytokine contents).   

Line 263 (Fig.4): In control group (G4), IL-4 contents is higher in heterozygous state of rs2243250 that in both homozygous genotypes, it should be  usually at intermediate level.    

Discussion

EBV is known to be in latent state in majority of population worldwide. Therefore, in cases of EBV seroconversion or PCR-based detection, one should tell about virus activation, probably, due to underlying HIV infection and its therapy, e.g., dysfunction of CD4+ lymphocytes and related cytokine effects.  Therefore, these factors should be clearly discussed and mentins as possible limitations of this extensive study. 

In general, the results shown in the manuscript are sometimes hard to understand because a number of studied parameters are presented in a non-systematized manner, despite extensive supplementary materials. In sum, the Results in main text should be set out in sequential and convincing way, with clear conclusions on the significant facts found in the study. Therefore, the manuscript should be sufficiently revised in order to determine clear aim of research, and to highlight the main facts and associations revealed in this extensive and multifacetted effort. The limitations of this study should be also listed (heterogeneity of HIV cohort, terms of preceding ARVT etc.).

Slight language editing is recommended.

Comments on the Quality of English Language

Minor copy editing

Author Response

Abstract:

Point 1. Line 21: The distinct aim of the study is not presented, description of clinical and laboratory parameters is listed instead.

Response 1: Thank you for your feedback. We have revised the abstract's objective for improved clarity. The inclusion of clinical and laboratory parameters is intended to provide a comprehensive understanding of how these SNPs influence the immune response in the context of these infections.

Point 2. The authors declare presence of differences …in all comparisons made in relation to the control group (G4)…. However, no results are present concerning potential differences except of expected differences, e.g., in CD4+ cells against control group. Among infected groups, only a difference of co-infected vs. mono-infected EBV is reported (for which parameters?). The results of SNP- immunity associations are not found, neither in Abstract, nor in Conclusion.

Response 2. Thank you for your feedback. The differences in all comparisons with the control group refer to various parameters, including plasma cytokine concentrations and CD4+/CD8+ T lymphocyte counts. Specifically, significant differences were observed in the plasma cytokine concentrations for three SNPs: rs2069762 (IL2) with higher IL-2 levels in HIV mono-infected individuals, rs2243250 (IL4) with higher IL-4 levels in the control group, and rs2069705 (IFNG) with higher IFN-γ levels in the co-infected group. We have now included these specific results in both the abstract (lines 26-37) and conclusion to provide a clearer understanding of the SNP-immunity associations. Once again, we thank you for your constructive suggestions, which have contributed to enhancing the quality of our work.

Materials and methods

Point 3. Line 103-105 and Table 1. One should provide data on mean (mediane) terms of HIV infection (years)  in  HIV patients, or duration of ARVT in these groups, because ARVT is a sufficient factor of immune changes as noted further in Discussion. The ARVT effects may sufficiently hide the genetic associations with SNP’s studied.

Response 3: Dear reviewer, the samples used in this study were collected when the HIV-infected patients were still not undergoing antiretroviral therapy (ARVT). This information can be found in “2.1. Sampling”.

Point 4. Line 156: SNPs genotyping: was it performed by original technique (e.g., primer design), or by commercial test system as mentioned (line 168).

Response 4. Dear reviewer, we conducted genotyping using a commercial kit, specifically the rhAmp™ SNP Genotyping kit, following the manufacturer's recommendations provided by Integrated DNA Technologies.

Results:

Point 5. Section 3.3: (SNPs and cytokine concentrations (line 241 and further on): The text description looks non-systematic and fragmentary, with a series of figures presenting some changes of different plasma cytokine contents in different study groups with SNP’s of appropriate cytokine genes. This design prevents demonstration of clear associations between SNP’s and probable immune effects (upon CD4+ cells and cytokine contents).   

Response 5. The aforementioned section of the study (section 3.3) assessed the potential association between different genotypes of the analyzed cytokine SNPs and their influence on plasma levels. The analysis was conducted within each group (G1, G2, G3, and G4), revealing statistically significant associations only for three SNPs: rs2069762 (IL2) within the HIV mono-infected group (G2), rs2243250 (IL4) within the control group (G4), and rs2069705 (IFNG) within the co-infected group (G1). A more detailed analysis of the different SNP genotypes and cytokine plasma concentrations should be conducted through gene transcription rate analysis, a methodology which was not performed in the present study. Additionally, there was an analysis of the association between different SNPs and the counts of adaptive immune cells, CD4+ and CD8+ T lymphocytes (section 3.4).

Point 6. Line 263 (Fig.4): In control group (G4), IL-4 contents is higher in heterozygous state of rs2243250 that in both homozygous genotypes, it should be  usually at intermediate level.

Response 6. In theory, the IL-4 content should be at an intermediate level in the heterozygous state (CT) of rs2243250 when compared to the homozygous genotypes (CC and TT). However, it is crucial to highlight that various interactions can modulate the levels of this cytokine according to genotype, as evidenced by data from the GTEx Portal (accessible at https://www.gtexportal.org/home/; genecode ID ENSG00000131437.15). It is worth noting that the GTEx samples for this gene were also collected from blood. The analysis of GTEx data reveals that the expression of the cytokine does not show as significant a discrepancy as the expression of CT, suggesting a more complex and multifaceted regulation.

Discussion

Point 7.EBV is known to be in latent state in majority of population worldwide. Therefore, in cases of EBV seroconversion or PCR-based detection, one should tell about virus activation, probably, due to underlying HIV infection and its therapy, e.g., dysfunction of CD4+ lymphocytes and related cytokine effects.  Therefore, these factors should be clearly discussed and mentions as possible limitations of this extensive study. 

Response 7. Dear Reviewer, we have chosen to add a paragraph to inform readers about the main limitations of the study. We appreciate your feedback and hope this will adequately improve our work.

Point 8. In general, the results shown in the manuscript are sometimes hard to understand because a number of studied parameters are presented in a non-systematized manner, despite extensive supplementary materials. In sum, the Results in main text should be set out in sequential and convincing way, with clear conclusions on the significant facts found in the study. Therefore, the manuscript should be sufficiently revised in order to determine clear aim of research, and to highlight the main facts and associations revealed in this extensive and multifacetted effort. The limitations of this study should be also listed (heterogeneity of HIV cohort, terms of preceding ARVT etc.).

Response 8. Dear Reviewer, we have chosen to add a paragraph to inform readers about the main limitations of the study. We appreciate your feedback and hope this will adequately improve our work.

Point 9: Slight language editing is recommended.

Response 9: Thank you for your valuable feedback. We appreciate your suggestion for slight language editing and will make the necessary adjustments accordingly.

Reviewer 2 Report

Comments and Suggestions for Authors

Manuscript ID: viruses-3024448

Title: Immunogenetic profile associated with patients living with HIV-1 and Epstein-Barr virus (EBV) in the Brazilian Amazon region.

Authors: Barros Costa et al.

 In this study, Barros Costa et al. present a work evaluating the presence of several SNPs for several cytokines in patients infected by HIV and/or EBV in Pará, Brazil and they describe their epidemiological profile.

Overall, this is a somewhat interesting manuscript. Nevertheless, the Introduction would have benefited from having details to provide context on both the situation of HIV infection and EBV infection  in this area of Brazil which could have been done by placing the first three paragraphs of the Discussion at the Introduction. The testing methods used seem reasonably considered. However, the Reviewer has some concerns regarding the selection of the control group. With 220 individuals available as controls, the Authors should have selected a smaller group but one that would be better matched to their study groups, two of which were significantly small. In this study, the Control group was not matched in age, marital status or in their sexual preference. Therefore, the Reviewer has concerns on how these factors may have affected the conclusions drawn from this study.   Another concern is that the HIV+/EBV- group is quite large compared to the other two study groups which the Reviewer feels may also affect the results. It is the Reviewer’s opinion that this manuscript would benefit from being re-worked and re-analyzed with a more properly selected Control group.

Comments on the Quality of English Language

Only minor English editing needed.

Author Response

  • The suggestion to reorganize the text and transfer the first three paragraphs of the discussion to the introduction in order to improve the context of the situation of HIV and EBV infection in Brazil was accepted and structured as indicated by the reviewer.
  • There is indeed a clear difference in the number of samples between the study groups, G1 (n= 79); G2 (n=322); G3 (n=34) and G4 (n= 220). However, this disparity can be explained by the way the samples were acquired, which occurred by spontaneous demand, in different locations and during the same period of time. Group 3 (EBV mono-infected), for example, had the lowest number of samples due to the low demand at the site during the collection period.

Reviewer 3 Report

Comments and Suggestions for Authors

This MNS titled “Immunogenetic profile associated with patients living with HIV-1 and Epstein-Barr virus (EBV) in the Brazilian Amazon region.” By Costa IB et al., aimed to evaluate the presence of SNPs rs2069762 (IL2), rs2243250 (IL4), rs1800795 (IL6), rs1800796 (IL6), rs2069705 (IFNG) and rs1799964 (TNF) from participants with HIV and/or EBV in the State of Pará; describe the epidemiological profile; and compare CD4+ and CD8+ T lymphocyte counts between four groups, including co-infection with HIV and EBV, mono-infection with HIV or EBV, in reference to people without infection. Explore how SNPs of immune cytokines associated with cytokine levels or CD4/CD8 T cell levels, and mono- or co-infection of HIV/EBV is novel. However, the data analysis was poorly designed, and presented. Some claims and conclusions lack data support.

Major comments

1.     Lack of SNPs results in the abstract.

2.     Figures 3-9: Have no clue how concentrations of cytokines or CD4/CD8 T cell counts were measured under each SNP. Please explain the methods.

3.     Some claims and conclusions lack data support.

3.1. Lines 343-344: The authors said that “plasma concentrations of IFN-γ were elevated compared to the control group”, but where is the data? There is no data comparing plasma concentrations of cytokines among study groups.

3.2. Lines 363-365: Authors claimed that “Findings from the present study demonstrated no statistical difference when comparing HIV mono-infected and HIV/EBV co-infected groups in relation to the cytokines IFN-γ and TNF-γ.” I do not see the data anywhere. There is no data comparing plasma concentrations of cytokines among study groups. 

3.3. Lines 380-383: Authors claimed that “In the context of SNPs and cytokine plasma levels, there was a significant association related to the SNP rs2069705 (IFNG) between the homozygous CC and heterozygous CT genotypes concerning the EBV/HIV co-infected group, where individuals with the homozygous CC genotype expressed higher levels of IFN-γ”, but I do not see the results anywhere in the manuscript.

3.4. Lines 391-395: Authors claimed that “present study, which unveiled a greater prevalence of the homozygous CC genotype across all examined groups. This prevalence ranged from 75.95% in the HIV-infected group to 85.29% in the EBV mono-infected group. Likewise, the current study identified a higher prevalence of the C allele in all groups”, but the results were not provided anywhere in the manuscript. Figure 5 only shows plasma IFN-γ concentrations under 3 SNPs within G4 (people with no infection). The authors did not compare frequencies of SNPs for any cytokine among study groups.

3.5. Lines 422-423: Authors claimed that “A significant association was observed between the SNP rs2069705 and the CD4+ T cell count in two groups (HIV mono-infected and control group).” But I only see the CD4 and SNP re2069705 data in G2 (HIV mono-infection) in Figure 6.

3.6. Lines 424-425: Authors claimed that “The CT heterozygous genotype for SNP rs2069705 was associated with lower CD4+ T cell counts in some HIV mono-infected individuals”, but it was the CC homozygous genotype for SNP rs2069705 that associated with lower CD4+ T cell counts as shown in Figure 6.

3.7. Lines 443-445: Authors claimed that “Notably, homozygous GG genotypes for rs1800796 (IL6) and rs1800795 (IL6) were associated with higher CD8+ T cell shedding in HIV mono-infected individuals”. But where is the data?

3.8. Lines 445-446: Authors claimed that “the heterozygous genotype TC for rs1799964 (TNF) was associated with higher CD8+ T cell counts in HIV/EBV co-infected and HIV mono-infected individuals.” But I can only see significance in G2 (HIV mono-infection) in Figure 7.

3.9. Lines 465-466: Authors concluded that “This study provides evidence that the SNP rs2069705 (IFN) is associated with an elevated levels in IFN-γ levels”. But no data supports this conclusion. Figure 5 only shows rs2069705 (IFN) TT had higher IFN-γ concentration in people without infection (G4). This conclusion is unclear with insufficient data support.

4.     The data analysis was poorly designed.

4.1. Cytokine concentrations should be compared among study groups to identify the effect of co- or mono- infection.

4.2. The frequency of cytokine SNPs should be compared among study groups to identify the association between SNPs and co- or mono-infection

Minor comments

Lines 56-57: Do you mean “evidenced by lower CD4 count”? Need to be clear.

Lines 59-60: It is unclear how lower EBV viral load is associated with therapy outcome. Please clarify.

Line 60: IM should be defined before using the abbreviation. Please define here not later at line 68.

Line 110: Please explain why these people with HIV were not treated? The Brazil has been a leader in HIV treatment and prevention in Latin America since 1996, when it began providing free access to HIV/AIDS treatment. The country's National STD and AIDS Program (NAP) has been known for its successful treatment initiatives. The NAP has challenged the conventional wisdom that prevention is more cost-effective than treatment and has provided free universal access to highly active antiretroviral therapy (HAART) for all people living with HIV/AIDS since 1996.

Line 120: “Sorological” should be “serological.”

Line 137: Please define SEAMB.

Table 2:

·       (n;%) should be [n (%)].

·       Please use “.” to replace “,” for decimal point, for example <0.01 not <0,01; 63 (79.7%) not 63 (79,7%).

Lines 246-249 & Figure 3: If the difference is significant as said in line 246, please annotate Figure 3.

Lines 249-249 & Figure 4: If the difference is significant as said in line 249, please annotate Figure 4.

Lines 252-255, Figure 5 & Table S1:

·       If the difference is significant as said in line 252, please annotate Figure 5.

·       Line 254 said G1, while Figure 5 and Table S1 say G4, please clarify.

Lines 272-273 & Figure 6: If the difference is significant as said in line 272, please annotate Figure 6.

Lines 280-281 & Figure 7: If the difference is significant as said in line 280, please annotate Figure 7.

Line 288: change Figure 08 to Figure 8.

Figure 8 and 9: Please annotate statistical significance in figures.

Line 316, 322: “individuals infected with HIV”, “HIV mono-infected”, “HIV/EBV co-infected” are not people-first languages. There are many more like this, e.g. line 333, line 335 and so on. Please check through the text, and use people-first language, e.g. “individuals living with HIV”, “individuals with HIV mono-infection”.

Line 329-331: Based on data shown in Figures 1 and 2, the group with co-infection had lower CD4 but higher CD8 count, thus the claim in lines 329-331 “lower T cell counts” is inaccurate.

Comments on the Quality of English Language

English is OK.

Author Response

Major comments

(Point 1)  Lack of SNPs results in the abstract.

Response 1: Thank you for your feedback. The appropriate corrections were inserted in the text of the manuscript, lines 23-37.

(Point 2) Figures 3-9: Have no clue how concentrations of cytokines or CD4/CD8 T cell counts were measured under each SNP. Please explain the methods.

Response 2: Thank you for your feedback. The statistical tests used are not actually shown in these figures, as can be seen in Figures 1 and 2. For the association of the “plasma cytokine concentration” and SNP variables, each individual in the study was distinguished and grouped according to the plasma value of each cytokine and their respective genotypes (homozygous and heterozygous) for each SNP analyzed. The same method was used for the CD4 and CD8+ T cell count variable.

(Point 3) Some claims and conclusions lack data support.

3.1. Lines 343-344: The authors said that “plasma concentrations of IFN-γ were elevated compared to the control group”, but where is the data? There is no data comparing plasma concentrations of cytokines among study groups.

Response 3: Thank you for your feedback. The data exists, but it was not really included in the article. Therefore, the excerpt mentioned by the reviewer has been removed from this manuscript as it does not form part of the scope of the article.

(Point 4) 3.2. Lines 363-365: Authors claimed that “Findings from the present study demonstrated no statistical difference when comparing HIV mono-infected and HIV/EBV co-infected groups in relation to the cytokines IFN-γ and TNF-γ.” I do not see the data anywhere. There is no data comparing plasma concentrations of cytokines among study groups.

Response 4: Thank you for your feedback. The data exists, but it was not really included in the article. Therefore, the excerpt mentioned by the reviewer has been removed from this manuscript as it does not form part of the scope of the article.

(Point 5) Lines 380-383: Authors claimed that “In the context of SNPs and cytokine plasma levels, there was a significant association related to the SNP rs2069705 (IFNG) between the homozygous CC and heterozygous CT genotypes concerning the EBV/HIV co-infected group, where individuals with the homozygous CC genotype expressed higher levels of IFN-γ”, but I do not see the results anywhere in the manuscript.

Response 5: Thank you for your feedback. There was an error in the text, the genotype associated with higher concentrations of IFN-γ was the TT homozygote related to the control group (G4), as shown in figure 5.

(Point 6) 3.4. Lines 391-395: Authors claimed that “present study, which unveiled a greater prevalence of the homozygous CC genotype across all examined groups. This prevalence ranged from 75.95% in the HIV-infected group to 85.29% in the EBV mono-infected group. Likewise, the current study identified a higher prevalence of the C allele in all groups”, but the results were not provided anywhere in the manuscript. Figure 5 only shows plasma IFN-γ concentrations under 3 SNPs within G4 (people with no infection). The authors did not compare frequencies of SNPs for any cytokine among study groups.

Response 6: Thank you for your feedback. As described above, there was an error in the text, in fact the genotype associated with higher concentrations of IFN-γ was the TT homozygote, as shown in figure 5. The data on the prevalence of genotypes and alleles does exist, but is not actually included in the manuscript, which is why it has been removed.

(Point 7) 3.5. Lines 422-423: Authors claimed that “A significant association was observed between the SNP rs2069705 and the CD4+ T cell count in two groups (HIV mono-infected and control group).” But I only see the CD4 and SNP re2069705 data in G2 (HIV mono-infection) in Figure 6.

Response 7: Thank you for your feedback. There was an error in the text, as described in figure 6 the association occurred between the SNP rs2069705 only for the HIV mono-infected group (G2) in relation to the CD4+ T cell count. The correction was made in the manuscript.

(Point 8) 3.6. Lines 424-425: Authors claimed that “The CT heterozygous genotype for SNP rs2069705 was associated with lower CD4+ T cell counts in some HIV mono-infected individuals”, but it was the CC homozygous genotype for SNP rs2069705 that associated with lower CD4+ T cell counts as shown in Figure 6.

Response 8: Thank you for your feedback. There really was a mistake in the writing of the text, as shown in Figure 6, the genotype associated with lower CD4+ counts was the homozygous CC genotype. The manuscript has been corrected.

(Point 9) 3.7. Lines 443-445: Authors claimed that “Notably, homozygous GG genotypes for rs1800796 (IL6) and rs1800795 (IL6) were associated with higher CD8+ T cell shedding in HIV mono-infected individuals”. But where is the data?

Response 9: Thank you for your feedback. There really was a mistake in the writing of the text. There was an association only between SNP rs1800796 (IL6) and CD8+ cell count, where the CC genotype was associated with higher CD8+ cell counts in the control group (G4). The appropriate correction has been made in the manuscript.

(Point 10) 3.8. Lines 445-446: Authors claimed that “the heterozygous genotype TC for rs1799964 (TNF) was associated with higher CD8+ T cell counts in HIV/EBV co-infected and HIV mono-infected individuals.” But I can only see significance in G2 (HIV mono-infection) in Figure 7.

Response 10: Thank you for your feedback. There really was a mistake in the writing of the text. As shown in Figure 7, the homozygous CC genotype was associated with higher CD8+ T cell counts and only for the HIV mono-infected group. The appropriate correction has been made in the manuscript.

(Point 11) 4.1. Cytokine concentrations should be compared among study groups to identify the effect of co- or mono- infection.

Response 11: Thank you for your feedback. As it was not within the scope of this manuscript, the analysis proposed by the reviewer was not included in the study. Specifically in this manuscript, we prefer to focus on genetic variations.

(Point 12) 4.2. The frequency of cytokine SNPs should be compared among study groups to identify the association between SNPs and co- or mono-infection

Response 12: Thank you for helpful suggestion. We performed the tests and presented the results in “3.3. SNPs and plasma concentration of cytokines IL-2, IL-4, IL-6, TNF-α, and IFN-γ”.

Minor comments

(Point 13) Lines 56-57: Do you mean “evidenced by lower CD4 count”? Need to be clear.

Response 13: Thank you for your feedback. The phrase “evidenced by lower CD4 count” refers to CD4+ T lymphocyte counts of less than 500 cells per mm³. this information was inserted into the manuscript.

(Point 14) Lines 59-60: It is unclear how lower EBV viral load is associated with therapy outcome. Please clarify.

Response 14: Thank you for your feedback. A study evaluating EBV viral load as a marker of lymphoma in AIDS patients showed that viral load decreased rapidly after the start of lymphoma therapy and remained undetectable, as described in reference 10.

(Point 15) Line 60: IM should be defined before using the abbreviation. Please define here not later at line 68.

Response 15: Thank you for your feedback. On line 73 of the manuscript there was the first reference to Infectious Mononucleosis with the use of the abbreviation.

(Point 16) Line 110: Please explain why these people with HIV were not treated? The Brazil has been a leader in HIV treatment and prevention in Latin America since 1996, when it began providing free access to HIV/AIDS treatment. The country's National STD and AIDS Program (NAP) has been known for its successful treatment initiatives. The NAP has challenged the conventional wisdom that prevention is more cost-effective than treatment and has provided free universal access to highly active antiretroviral therapy (HAART) for all people living with HIV/AIDS since 1996.

Response 16: Thank you for your feedback. Samples from HIV-infected individuals were taken shortly after the patient was diagnosed with the infection, so they were free of antiretroviral therapy (ART) and then sent to the referral center for treatment.

(Point 17) Line 120: “Sorological” should be “serological.”

Response 17: Thank you for your feedback. Correction made to the text.

(Point 18) Line 137: Please define SEAMB.

Response 18: Thank you for your feedback. SEAMB corresponds to a scientific section within the Evandro Chagas Institute where the cytometer used to measure the plasma concentration of the cytokines in the study is located. As this is not relevant information for the article, we decided to remove it from the text.

(Point 19): Table 2: (n;%) should be [n (%)].

Response 19: Thank you for your feedback. The appropriate correction has been made.

(Point 20): Please use “.” to replace “,” for decimal point, for example <0.01 not <0,01; 63 (79.7%) not 63 (79,7%).

Response 20: Thank you for your feedback. The appropriate correction has been made.

(Point 21) Lines 246-249 & Figure 3: If the difference is significant as said in line 246, please annotate Figure 3.

Response 21: Thank you for your feedback. The statistically significant difference was represented in the figure, as proposed by the reviewer.

(Point 22) Lines 249-249 & Figure 4: If the difference is significant as said in line 249, please annotate Figure 4.

Response 22: Thank you for your feedback. The statistically significant difference was represented in the figure, as proposed by the reviewer.

(Point 23) Lines 252-255, Figure 5 & Table S1: If the difference is significant as said in line 252, please annotate Figure 5.

Response 23: Thank you for your feedback. The statistically significant difference was represented in the figure, as proposed by the reviewer.

(Point 24) Line 254 said G1, while Figure 5 and Table S1 say G4, please clarify.

Response 24: Thank you for your feedback. The correct group refers to the control group (G4). The appropriate correction has been made in the text.

(Point 25) Lines 272-273 & Figure 6: If the difference is significant as said in line 272, please annotate Figure 6.

Response 25: Thank you for your feedback. The statistically significant difference was represented in the figure, as proposed by the reviewer.

(Point 26) Lines 280-281 & Figure 7: If the difference is significant as said in line 280, please annotate Figure 7.

Response 26: Thank you for your feedback. The statistically significant difference was represented in the figure, as proposed by the reviewer.

(Point 27) Line 288: change Figure 08 to Figure 8.

Response 27: Thank you for your feedback. The appropriate correction made.

(Point 28) Figure 8 and 9: Please annotate statistical significance in figures.

Response 28: Thank you for your feedback. The statistically significant difference was represented in the figure, as proposed by the reviewer.

(Point 29) Line 316, 322: “individuals infected with HIV”, “HIV mono-infected”, “HIV/EBV co-infected” are not people-first languages. There are many more like this, e.g. line 333, line 335 and so on. Please check through the text, and use people-first language, e.g. “individuals living with HIV”, “individuals with HIV mono-infection”.

Response 29: We understand the reviewer's concern with the most appropriate nomenclature in relation to individuals living with HIV. The suggested adjustments have been made.

(Point 30) Line 329-331: Based on data shown in Figures 1 and 2, the group with co-infection had lower CD4 but higher CD8 count, thus the claim in lines 329-331 “lower T cell counts” is inaccurate.

Response 30: This statement refers to the CD4+ T-lymphocyte count. The appropriate correction has been inserted in the text of the manuscript.

Round 2

Reviewer 1 Report

Comments and Suggestions for Authors

The considered article was subject to some amendments in response to reviewers. Most required corrections are done

Clear aims of the study still are not formulated, but a long fragment of Results is included, thus explaining the scope of study.

Detailed epidemiological data (on the duration of disease and ARVT terms) are, however, not included into Materials and Methods, may be, due to mangling information.

The limitations of this study are listed more distinctly at the end of Discussion.

EBV reactivation, probably resulting from HIV-induced immune deficiency is clearly mentioned in conclusion.

In general, the article may be published, taking into account some deficiency of clinical/epidemiological data.

Comments on the Quality of English Language

Minor editing of amended fragments is required.

Author Response

Dear,
Thanks for the helpful comments. The suggested changes were included in the new version of the manuscript.

Reviewer 2 Report

Comments and Suggestions for Authors

Manuscript ID: viruses-3024448

Title: Immunogenetic profile associated with patients living with HIV-1 and Epstein-Barr virus (EBV) in the Brazilian Amazon region.

Authors: Barros Costa et al.

 In this study, Barros Costa et al. present a work evaluating the presence of several SNPs for several cytokines in patients infected by HIV and/or EBV in Pará, Brazil and they describe their epidemiological profile.

 In response to first review, the Authors incorporated the suggested changes to the Introduction and now provide adequate background information. Additionally, some other areas of the manuscript were also re-worked. Nevertheless, the manuscript in general, requires proofreading to correct sentences and remove repeated words and phrases, for example Lines 82-83.  One point the Authors did answer was the Reviewer’s question about the Control group regarding to why the control group was not matched to the demographics, age, marital status or sexual preference of the study groups and how this could affect the results and conclusions drawn from this study.

The Results section was revised; however, in this process the manuscript leads to confusion regarding the results due to the inconsistency in the labelling of the study groups both in the text and figures.

·       In Lines 284-286, The difference occurred in relation to the co-infected group 284 (G1) (G4), where it was observed that the homozygous TT genotype expressed higher 285 plasma levels of IFN-γ when compared to the other genotypes (Figure 5)”; it is un-clear whether they refer to the group “co-infected” or (G-1) as originally labelled or to the “control” group (G-4).

·       In Line 321, The EBV+/HIV- group (G2) demonstrated a significant difference for the 321” the Authors marks it as (G-2) however according to the manuscript the group (G-2) refers to the “HIV mono-infection” and the EBV+/HIV- group should be marked as (G3).

·       In Figure 7, it is unclear whether it is for group (G-2) or “HIV mono-infected” or if it refers to the EBV+/HIV- group which is group (G3).

In the Discussion section, revision is required:

·       In Lines 463-473, there are errors in terminology (i.e. homozygous vs. heterozygous), incomplete sentences or sentences that make no sense as to  their meaning.

·       In Lines 390-394, should change the abbreviation “TARV” to the English abbreviation “ART”.

In the Conclusion, Lines 521-523, the sentence should be revised for clarity and removing repeated words.

The Reviewer suggests a thorough proofreading of the entire manuscript to verify text clarity and sentence form and to ensure consistency in labelling of groups in both text and figures. 

Comments on the Quality of English Language

Some English editing is required. There are repeated words in sentences, incomplete sentences and sentences that are un-clear (either need re-phrasing or there are sintax errors present). 

Author Response

(The authors gave the same response as above.)
